# Repeatedly experiencing the McGurk effect induces long-lasting changes in auditory speech perception

John F. Magnotti [1], Anastasia Lado[1], Yue Zhang [2], Arnt Maasø [3], Audrey Nath[4] & Michael S. Beauchamp [1] ✉

In the McGurk effect, presentation of incongruent auditory and visual speech evokes a fusion percept different than either component modality. We show that repeatedly experiencing the McGurk effect for 14 days induces a change in auditory-only speech perception: the auditory component of the McGurk stimulus begins to evoke the fusion percept, even when presented on its own without accompanying visual speech. This perceptual change, termed fusion-induced recalibration (FIR), was talker-specific and syllable-specific and persisted for a year or more in some participants without any additional McGurk exposure. Participants who did not experience the McGurk effect did not experience FIR, showing that recalibration was driven by multisensory prediction error. A causal inference model of speech perception incorporating multisensory cue conflict accurately predicted individual differences in FIR. Just as the McGurk effect demonstrates that visual speech can alter the perception of auditory speech, FIR shows that these alterations can persist for months or years. The ability to induce seemingly permanent changes in auditory speech perception will be useful for studying plasticity in brain networks for language and may provide new strategies for improving language learning.

Viewing the face of a talker powerfully influences the perception of auditory speech, as exemplified by the McGurk effect, an illusion in which incongruent auditory and visual speech evokes a fusion percept different from either component modality (McGurk and MacDonald, 1976). The McGurk effect has been influential in the development of computational models of multisensory integration, as it presents a prime example of causal inference: the incongruence between face and voice means that the observer must infer the likelihood that the face and voice were caused by the same talker or by different talkers, and adjust perception accordingly[1–5].

Demonstrating the McGurk effect is simple: listen to the unambiguous auditory speech that constitutes that auditory component of a McGurk stimulus; listen to the same speech while viewing the talker's face; marvel at the perceptual difference. Author AM's demonstration of the McGurk effect has received more than 800,000 views on YouTube[6]. After many years of demonstrating the McGurk effect using AM's stimulus, authors JFM and MSB noticed a striking change. There was no longer any difference between listening to the speech on its own and listening while viewing AM's face: both conditions evoked the illusory McGurk fusion percept. This dramatic change in perception persists to the present day, many years later. Perceptual

changes that last for a few seconds following McGurk exposure have been studied for decades and are thought to reflect phonetic recalibration driven by prediction error[7–10]. This led us to coin the term fusion-induced recalibration (FIR) to describe McGurk-induced perceptual changes that can last for years. The *Discussion* examines the relationship between FIR and other cross-modal changes in auditory perception in detail.

The purpose of the present investigation was to determine whether FIR could be induced in naive participants, and, if so, to construct a causal inference model of the process. These questions are pertinent because long-lasting changes in auditory speech perception induced by visual speech could provide a useful tool for examining the neural basis of changes in auditory speech representations and aid in the development of therapies for patients struggling with speech perception or language learning.

Previous studies presented AM's McGurk stimulus in a single testing session or two testing sessions one year apart[11,12]. No long-lasting aftereffect was observed in these studies, suggesting that more frequent exposure was necessary. Therefore, in the current study, participants were briefly exposed to AM's McGurk stimulus every day for 14 days, as a compromise between a single exposure session (ineffective based on previous studies) and exposing

[1]Department of Neurosurgery, Perelman School of Medicine, University of Pennsylvania, Philadelphia, PA, USA. [2]Department of Neurosurgery, Baylor College of Medicine, Houston, TX, USA. [3]Institute for Media and Communications, University of Oslo, Oslo, Norway. [4]Department of Neurosurgery, University of Texas Medical Branch, Galveston, TX, USA. ✉e-mail: beaucha@upenn.edu

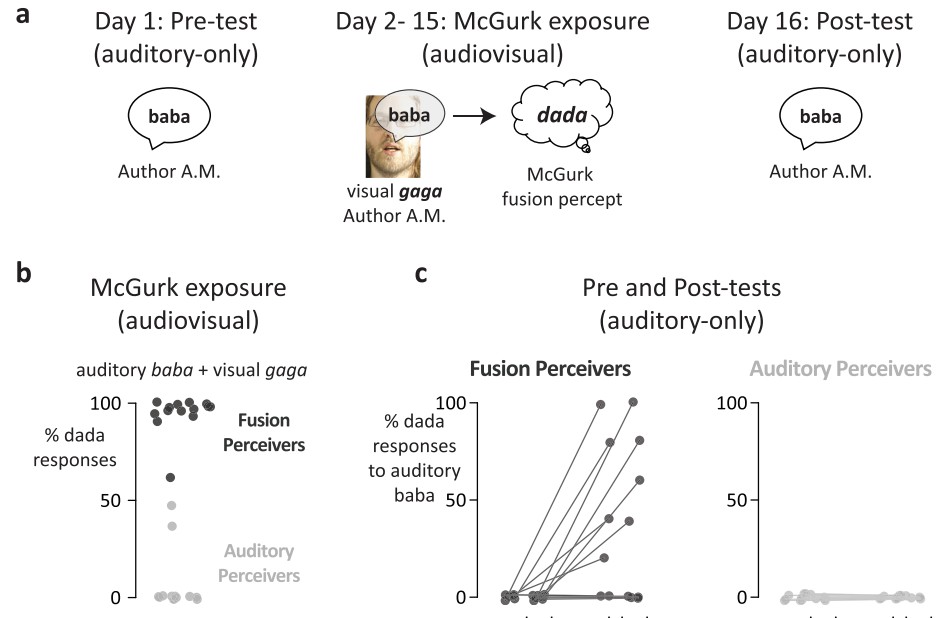

**Fig. 1 | Fusion-induced recalibration. a** On the first day of the experiment, participants reported their perception of an unambiguous, auditory-only *baba* spoken by author AM, randomly intermixed with control syllables. On each of the next 14 days (McGurk exposure), participants reported their perception of AM's *baba* paired with AM's incongruent visual *gaga*, expected to induce the McGurk fusion percept of *dada*. Five repetitions were presented on each day, randomly intermixed with other stimuli. On Day 16, participants reported their perception of AM's auditory-only *baba*, randomly intermixed with control syllables. **b** During McGurk exposure, half of the participants frequently reported the fusion percept of *dada* (fusion perceivers; one black circle per participant; *n* = 14 participants). The remaining participants (*n* = 14) did not experience the McGurk effect. Instead, they usually perceived *baba*, the auditory component of the McGurk stimulus (auditory perceivers; one gray circle per participant). In this plot and all subsequent plots, symbols are jittered along the horizontal axis to enhance visibility. **c** Fusion perceivers showed changes in perception of the auditory component of the McGurk stimulus (auditory-only *baba*) from the pre-test to the post-test. In the pre-test, perception was veridical (participants always perceived *baba*), but in the post-test, fusion perceivers often reported a non-veridical percept of *dada*. In contrast, auditory perceivers perceived *baba* veridically in both the pre-test and the post-test (zero *dada* reports). Only fusion perceivers showed a perceptual change, prompting the term fusion-induced recalibration.

participants frequently for years (effective based on the experience of the authors but experimentally intractable). On each of the fourteen exposure days, participants viewed five repetitions of AM's McGurk stimulus (about 10 s in total), comparable to the authors' daily exposure in the course of teaching and research.

Before exposure, naive participants were expected to accurately perceive A.M.'s auditory speech and control speech. The experimental hypothesis was that following brief daily exposure to AM's McGurk stimulus, participants would no longer accurately perceive AM's auditory speech: instead, they would experience the McGurk fusion percept. Perception of control speech was expected to be unchanged. No feedback was ever given to prevent any response bias due to demand characteristics[13].

## Methods
There was no preregistration of any study. All relevant ethical regulations were followed, and all experiments were approved by the Institutional Review Board (IRB) of the University of Pennsylvania. Because participants were tested online, a waiver for written informed consent was obtained from the IRB. Participants were compensated as approved by the IRB, at an approximate hourly rate of $15.

## Overview
The hypothesis for the study was that repeated exposure to audiovisual McGurk stimuli would alter the perception of the auditory-only component of the McGurk stimuli. As shown in Fig. 1, the design consisted of an auditory-only pre-test on Day 1; brief exposure to audiovisual McGurk stimuli on Days 2–15; and an auditory-only post-test conducted on Day 16, at least 24 h following the final audiovisual exposure. Additional auditory-only post-tests were conducted up to 1 year following the final audiovisual exposure. Data and code are freely available[14].

## Participants
All data were collected online from participants using the SoSci Survey research platform[15]; previous studies of the McGurk effect showed similar results for in-person and online participants[11]. Thirty participants completed the pre-test. Two participants performed poorly and were excluded from further analysis, leaving 28 participants.

The mean age was 26 years (range 19–43). Participants self-identified as "Male" (*n* = 17), "Female" (*n* = 11), or "Other" (*n* = 0). Data on race and ethnicity was not collected. All participants affirmed that they did not have a hearing impairment that would make it difficult to understand words; that they did not have an uncorrected visual impairment that would make it difficult to watch a video of a person talking; and that they were able to either complete tasks in a quiet environment or use noise-canceling headphones.

## McGurk stimuli
The primary McGurk stimulus (*S1*) consisted of the McGurk stimulus recorded by male author AM, consisting of auditory *baba* paired with visual *gaga* (expected fusion percept of *dada*.) *S1* was responsible for the original observation by the authors. An additional McGurk stimulus (*S2*) was recorded by female author AN and consisted of auditory *ba* paired with visual *ga* (fusion percept of *da*).

## Auditory stimuli
The auditory stimuli consisted of the auditory component of *S1* (auditory *baba*; *S1_Aud*); the auditory component of *S2* (auditory *ba*; *S2_Aud*); and 25 control stimuli, all from different talkers: 13 different *ba* stimuli (five different F talkers; eight different M talkers); two different *baba* stimuli (1 F, 1 M); five different *da* stimuli (2 F; 3 M); five different *ga* stimuli (3 F; 2 M).

## Experimental design

In the pre-test and the post-test, participants reported their perception of auditory-only stimuli: five repetitions of *S1_Aud* and ten repetitions of *S2_Aud* (this equated the total number of syllables, since *S1* contained two syllables while *S2* was monosyllabic) along with one repetition of each of the control stimuli. Stimulus order was randomized, with a different random order (but the exact same stimuli) in the pre-test and the post-test. Randomization ensured that the multiple repetitions of *S1_Aud* and *S2_Aud* were not presented in succession, avoiding any possible verbal transformation effects.

On each of the fourteen McGurk exposure days, participants reported their percept of five repetitions of *S1* and ten repetitions of *S2*, randomly interleaved (due to an experimental error, only nine repetitions of *S2* were presented on Days 8 through 13).

## Response collection and coding

Following every stimulus presentation, respond to the prompt "type what you think the person said in the box below" with an open-choice response text box. No feedback was ever given. Responses were coded, blind to the stimulus that elicited the response, into one of four categories: *Auditory*, *Visual*, *Fusion*, and *Other*. Responses to auditory-only and McGurk stimuli were coded identically (e.g., a response of *da* to auditory-only *ba* was scored as a *Fusion* response.) Responses were coded based on the initial consonant of the response (e.g., *buh buh* for *baba* was classified as *Auditory*).

## Mixed-effects models for behavioral data

All data were analyzed using R (Computing, 2022). Coded responses were analyzed using binomial-family generalized linear-mixed effects models (GLMEs) as implemented in the *lme4* package (Bates et al., 2015). Mixed-effects models were used because they provide a consistent approach to understanding the effect of both categorical and numeric independent variables (fixed effects) while explicitly modeling other sources of variation (random effects such as participant effects or stimulus effects).

The dependent variable for all statistical tests consisted of the count of fusion responses vs. non-fusion responses. Binomial models (function *glmer* with the family set to *binomial*) were used because the data was not normally distributed. All models contained a random effects term for the participant; if the fit was singular, the nested relationship was replaced with just an intercept at the participant level. Models with multiple stimuli also included a random intercept term to account for variation across stimuli. Statistical significance of main effects and interactions was tested using Analysis of Deviance, with Type II sums of squares as implemented in the *car* package[16]. Post hoc tests were implemented using the *emmeans* package. The odds-ratio comparing the likelihood of fusion response vs. non-fusion response between conditions is used as the measure of effect size, alongside 95% confidence intervals (*emmeans* calculates the confidence intervals in the log-odds space then back-transforms them into the original space). Confidence intervals for the correlation coefficient were calculated in Z space (using Fisher's r-to-z transform) before back-transforming into the original space, as implemented in the *confintr* package.

## Long-term follow-up tests

Long-term auditory-only tests (identical to the pre-test and post-test but with different randomizations) were given approximately every 30 days. The timing of each long-term post-test was determined relative to the final audiovisual exposure day. For instance, the first long-term post-test was administered 30 days after the final audiovisual exposure day (equivalent to 46 days since the pre-test). By the sixth long-term post-test (180 days after the final audiovisual exposure day and 196 days after the pre-test), many participants did not show any changes in auditory perception, and long-term testing was discontinued for these participants. For the remaining participants, testing continued.

To assess the change in the percentage of fusion responses over the long-term follow-up tests, we used non-linear least squares (NLS) models (function *nls* from the R *stats* package) of the form: *fusion* = $a * e^b$. This

formulation estimates a proportional change in the percentage of fusion responses, starting at some initial value at the post-test (the *a* term in the model) and changing at a rate of $e^b$ (decaying if $b < 0$) each month.

Visual inspection showed three distinct groups of participants: participants with fusion responses that persisted unchanged over time; participants whose fusion responses decayed to zero over the first few long-term testing sessions; and participants who never had any fusion responses. Separate NLS models were fit for each group. Statistical significance of the overall model was determined by comparing the residuals between an intercept-only model (no change over time) and the intercept + slope model (using *anova* from the *stats* package).

## Replication experiment

A replication experiment was conducted using *S1* (male author AM's auditory *baba* paired with visual *gaga*, expected fusion percept of *dada*) and the other syllable pairing described by McGurk and MacDonald, auditory *pa* paired with visual *ka*, fusion percept of *ta*[17]. This stimulus (termed *S3* to avoid confusion with *S2* in the main experiment) consisted of female author AN's auditory *pa* paired with visual *ka*, expected fusion percept of *ta*.

Thirteen participants were tested using Amazon MTurk. The design was identical to the main experiment. For the pre-test, participants were presented with ten repetitions each of *S1_Aud* and *S3_Aud* and 20 total control stimuli, five repetitions each of four stimuli: *baba* (2 M); *pa* (2 F). On each of the 14 exposure days (the same duration as the original study), participants were presented with ten repetitions each of *S1_AV* and *S3_AV*. The post-test was conducted with the same stimuli as the pre-test but in a different random order (within each phase, stimuli were randomly interleaved).

To simplify data analysis, forced-choice responding was used, rather than the open-choice responding in the main experiment. Participants were instructed to answer the question "What did the person say?" by selecting one response from the following list: 'BaBa', 'GaGa', 'DaDa', 'Pa ', 'Ka', 'Ta', 'ThaTha', 'FaFa', 'Other'. During the pre-test and post-test, participants responded after every stimulus. During the exposure days, participants only responded following the final repetition of *S1_AV* or *S3_AV*. No feedback was ever given. In the exposure phase, one participant completed only 12 days of training; four participants completed only 13 days of training; the remaining eight participants completed all 14 days of training.

## Computational model construction and fitting

The CIMS model was fit as previously described in refs. [1],[3]. A single representational space was used across participants; *da* is located between *ba* and *ga* in the space[18],[19]. The encoding distributions were fixed across participants. Visual features are mapped to the y-axis and auditory features to the x-axis, with the result that an auditory token has more uncertainty along the y-axis, and a visual token has more uncertainty along the x-axis. The Bayesian combination of A and V produces a pooled representation midway intermediate to the A and V representations. Causal inference is used to create a final AV representation that is the combination of the A representation and the pooled representation, weighted by the observer's estimation that the auditory and visual speech arise from the same talker ($C = 1$) or different talkers ($C = 2$); only the final, integrated AV representation is shown in the figures. On each trial, a location within the encoding distribution is generated and used to construct the final AV representation; the distance between the A and final AV representations is a measure of the error signal. The new auditory representation (A') is located on a line between the original auditory representation (A) and the final AV representation, with the distance along the line proportional to the number of McGurk percepts on the first audiovisual exposure day. This resulted in variable A' for different participants, with A' = A for participants who did not perceive the McGurk effect.

For the 24-h post-test, individual trials were simulated by selecting locations within their modeled A', and then a perceptual response was generated based on the encoded location. This produced a model estimate of

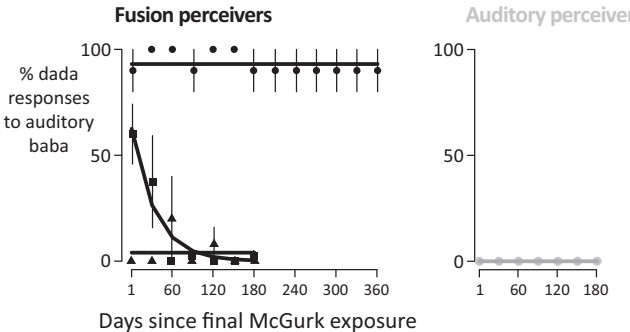

**Fig. 2 | Time course of fusion-induced recalibration.** Participants ($n = 26$) completed additional auditory-only post-tests at monthly intervals, without any additional McGurk exposure. For participants who experienced the McGurk effect during the 14 days of McGurk exposure (fusion perceivers), there were three patterns of responses. One group of participants reported many *dada* percepts at every post-test (circles; $n = 2$). A second group reported a moderate number of *dada* percepts at the first post-test, and fewer at subsequent tests (squares; $n = 5$). A third group reported few *dada* percepts at any time point (triangles; $n = 6$). For participants who did not experience the McGurk effect during exposure (auditory perceivers), there were no *dada* percepts at any time point (gray circles; $n = 13$). Error bars show the standard error of the mean across participants. Lines show the best-fit non-linear regression model for each group.

% FIR percepts for each participant. To assess model fit, the model predictions were correlated with the actual % FIR for each participant. For visualization in Fig. 2d, the standard error of the model and the participants was estimated. For the model, the mean fit was used to estimate randomly generated binomial data: the mean fit assumes thousands of trials, while each individual participant only had ten trials. For participants, the pooled FIR variance was estimated across all levels of McGurk, weighted by the number of participants at each level. The SEM was estimated as the SD of the pooled variance divided by the square root of the number of participants at that level.

### Reporting summary

Further information on research design is available in the Nature Portfolio Reporting Summary linked to this article.

## Results

### Bimodal distribution of McGurk susceptibility

Every day for 14 days, participants reported their perception of five repetitions of AM's McGurk stimulus, consisting of auditory *baba* paired with visual *gaga*. Consistent with previous research[11], there was a bimodal distribution of susceptibility to the McGurk effect (Fig. 1b). Half of the participants ($n = 14$) frequently experienced the McGurk effect, reporting the fusion percept of *dada* in 95% of trials; these participants were termed *fusion perceivers* for AM's McGurk stimulus. The remaining participants ($n = 14$) rarely or never experienced the McGurk effect (fusion percept on 6% of trials), instead primarily perceiving the auditory component of the stimulus (*baba*); these participants were termed *auditory perceivers* for this stimulus.

### Changes in auditory perception in fusion perceivers but not auditory perceivers

Fusion perceivers showed a change in auditory perception from the pre-test to the post-test. In the pre-test, the perception of AM's auditory-only *baba* was veridical: participants responded *baba* on 99% of trials and never responded *dada*. In contrast, the post-test perception was not veridical: participants reported *dada*, the percept expected from fusion-induced recalibration (FIR), in 37% of trials. Auditory perceivers (who did not experience the McGurk effect during exposure) showed no changes in auditory perception from the pre-test to the post-test: *baba* was perceived veridically as *baba*, never as *dada*.

A generalized linear-mixed effects model (GLME) was fit to the data with the dependent variable set to the number of *dada* responses; fixed factors of time (pre-test *vs.* post-test), and perception during audiovisual exposure (fusion perceiver *vs.* auditory perceiver); and a random effect of the participant. The model yielded significant main effects of both time ($\chi^2_1 = 23$, $p = 10^{-6}$) and McGurk perception ($\chi^2_1 = 11$, $p = 0.0009$), along with an interaction between time and McGurk perception ($\chi^2_1 = 17$, $p = 10^{-5}$) driven by the different post-test reports of fusion perceivers and auditory perceivers. Post hoc tests confirmed a significant increase in *dada* reports from pre-test to post-test for fusion perceivers (0% to 41%, $z = 6$, $p < 10^{-10}$; odds-ratio: 8.5, 95% confidence interval (CI): 4.4 to 16.6) but no change for auditory perceivers (0% to 0%, $z = 0$, $p \sim 1$; odds-ratio = 1.0, 95% CI: 0.5, 2.2).

### Long-term follow-up: auditory-only

To examine the persistence of the perceptual change, the perception of AM's *baba* was examined in the months following the 14-day exposure period, without any additional audiovisual McGurk exposure (Fig. 2). These long-term post-tests were identical to the auditory-only pre- and post-tests.

Thirteen fusion perceivers (participants who perceived the McGurk effect for AM's stimulus during audiovisual exposure days) completed long-term testing (Fig. 2). Three distinct response patterns were observed. One group of fusion perceivers ($n = 2$ out of 13) reported many FIR responses at the 24-h post-test (mean of 90%) and continued to frequently report the FIR percept at subsequent post-tests, up to and including 360 days following the final McGurk exposure. Another group ($n = 5$) reported a moderate number of FIR responses at the 24-h post-test (mean of 60%) and decreasing numbers of FIR percepts at subsequent post-tests (decay time constant of 42% per month). A third group of fusion perceivers ($n = 6$) reported few FIR percepts (mean of 1%) at any time point. In auditory perceivers (who did not experience the McGurk effect during exposure days), no FIR responses were ever reported.

### Control stimuli

Control syllables were also presented in the auditory pre-test and post-test, consisting of a variety of syllables spoken by different talkers. Accuracy was high and unchanged between the pre-test (95%) and the post-test (97%), demonstrating that the results cannot be explained by response bias, such as responding *dada* to every stimulus.

Two control stimuli consisted of auditory *baba* spoken by male talkers other than AM. While the perception of AM's *baba* changed following exposure to AM's McGurk stimulus, perception of the control *baba* stimuli did not (0% *dada* responses in both the pre-test and the post-test), demonstrating that the perceptual change from *baba* to *dada* did not generalize from AM to other talkers.

### Additional McGurk stimulus

Another McGurk stimulus from a different talker (*S2*) was presented to participants using the same procedure. *S2* was selected because, in previous experiments, it was much less effective than *S1* (AM's *baba*/gaga) at eliciting the McGurk fusion percept[11,12] allowing exploration of whether mere *exposure* to a McGurk stimulus was sufficient to alter perception or whether actually perceiving the McGurk fusion percept was required. As expected, *S2* was less effective than *S1*: only four participants were *S2* fusion perceivers, compared with fourteen for *S1*. FIR was observed in two of the four *S2* fusion perceivers, with 5% fusion responses to *S2_Aud* in the pre-test vs. 40% in the post-test. Seven participants showed FIR for *S1* but not *S2* despite identical exposure to both stimuli.

### Replication

The initial experiment investigated FIR using the pairing of auditory *ba* and visual *ga* (*S1* and *S2*). The McGurk effect also occurs for another syllable pairing, that of auditory *pa* and visual *ka*, with an expected fusion percept of *ta*[17]. To determine whether FIR could be induced by auditory *pa* and visual *ka*, the same experimental design used in the initial experiment was repeated

**Fig. 3 | Replication of FIR with two stimuli in a different group of participants.** In a replication experiment, a different group of participants (*n* = 13) was tested using the experimental paradigm shown in Fig. 1a. **a** During exposure to author AM's McGurk stimulus (auditory *baba* and visual *gaga*, expected fusion percept of *dada*) *n* = 6 participants frequently reported *dada* (fusion perceivers; one black circle per participant). The remaining participants (*n* = 7) perceived *baba*, the auditory component of the McGurk stimulus (auditory perceivers; gray circles). **b** Fusion perceivers of the AM McGurk stimulus showed a perceptual change for AM's auditory-only *baba* from the pre-test to the post-test, with frequent reports of *dada* in the post-test. In contrast, auditory perceivers perceived *baba* veridically (few *dada* reports) in pre- and post-tests. **c** During exposure to female author AN's McGurk stimulus consisting of auditory *pa* and visual *ka*, expected fusion percept of *ta*, *n* = 10 participants frequently reported the fusion percept while the remaining *n* = 3 participants did not. **d** For fusion perceivers of the AN McGurk stimulus, perception of the auditory component of the stimulus (auditory *pa*) changed from the pre-test to the post-test, with more frequent *ta* percepts. For auditory perceivers of the AN McGurk stimulus, perception was unchanged.

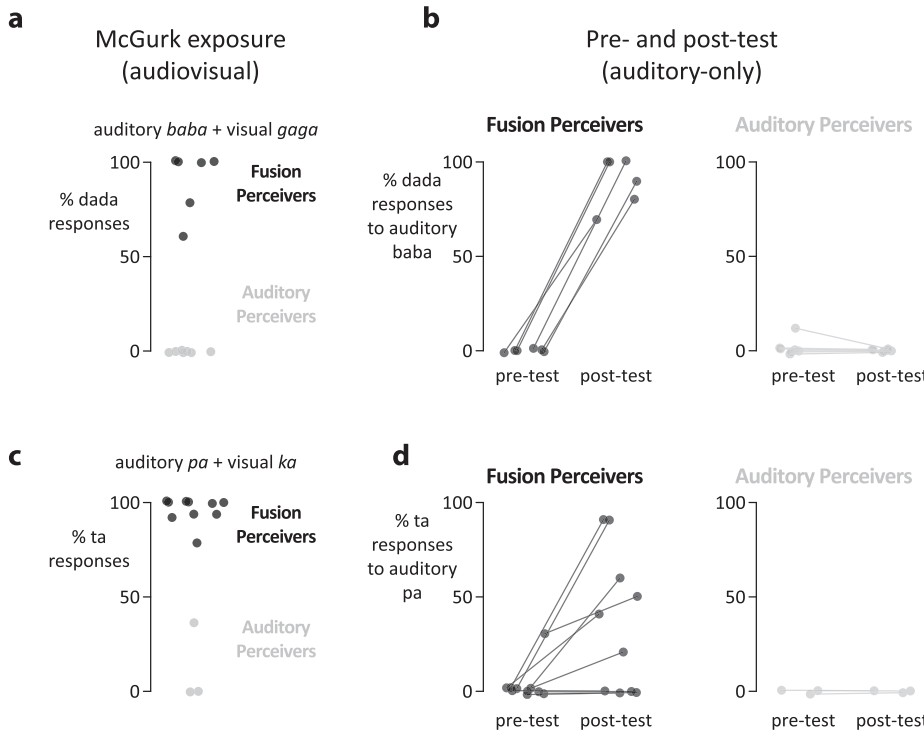

with a second group of participants (*n* = 13; Fig. 3). The hypothesis was that repeated exposure to this pairing (S3; female author AN's recording of auditory *pa* with visual *ka*) would induce a change in the perception of AN's *pa* to *ta* in the auditory-only post-test. As a replication, AM's *baba*/*gaga* McGurk stimulus (S1) was also tested.

In the auditory-only pre-test, participants accurately identified AM's *baba* (responding *baba* on 98% of trials) and AN's *pa* (responding *pa* on 96% of trials). As expected, there was a bimodal distribution of responses to both McGurk stimuli. For stimulus, 6 of 13 participants frequently experienced the McGurk effect, reporting the fusion percept of *dada* on 90% of trials, while seven participants rarely experienced the effect (fusion percept on 0% of trials). For AN's stimulus, ten participants frequently experienced the McGurk effect (fusion percept of *ta* on 96% of trials), while three participants did not (12% fusion percepts).

Changes in auditory perception were observed, but only in participants who experienced the McGurk effect for a stimulus (fusion perceivers). For participants who experienced the McGurk effect for AM's stimulus, *baba* was perceived as *dada* on 90% of trials in the post-test, compared with 0% in the pre-test ($X^2_1 = 57$, $p = 10^{-14}$; odds-ratio 22, 95% CI: 9.9, 49.1). For participants who experienced the McGurk effect for AN's stimulus, *pa* was perceived as *ta* on 35% of trials in the post-test, compared with 3% of trials in the pre-test ($X^2_1 = 18$, $p = 10^{-5}$; odds-ratio 3.5, 95% CI: 2.0, 6.3).

**Computational model**

The causal inference of the multisensory speech (CIMS) model provides a principled explanation for the perception of McGurk stimuli and other incongruent audiovisual speech[1,3]. The model uses a two-dimensional space to characterize the auditory and visual information received by the observer during speech perception. During the presentation of audiovisual speech, multiple internal representations are constructed: one for the auditory speech, one for the visual speech, and one for the integrated (audiovisual) representation. The integrated representation uses causal inference to combine the auditory and visual representations, taking into account the likelihood that the voice and face were caused by the same talker. To account for FIR, the CIMS model was revised to incorporate multisensory prediction error[19,20].

Figure 4 illustrates the CIMS model of FIR for the McGurk pairing of auditory *ba* with visual *ga* with an expected fusion percept of *da*. We consider three different participants who vary in their susceptibility to the McGurk effect. For a participant who frequently perceives the McGurk effect (Fig. 4a), the integrated (audiovisual) representation lies in the *da* region of representational space because of their high likelihood of assigning a common cause to the incongruent auditory and visual speech. Across trials of an identical McGurk stimulus, the representations of auditory and audiovisual representations differ on each trial due to sensory noise (Fig. 4b) but there is a consistent offset in their locations in representational space. To eliminate this error, the auditory representation, initially centered in the *ba* region of representational space, shifts towards the audiovisual representation in the *da* region of representational space. The shift is induced by the McGurk fusion percept that arises from the integrated audiovisual representation. The shift can be measured during the auditory-only post-test (Fig. 4c). Across trials, noisy encoding means that there is variability in the representation of the *ba* auditory stimuli, but it always lies in the *da* region of representational space, so the participant always reports the *da* percept, corresponding to a high degree of FIR. This result models the group of participants shown in Fig. 2 who reported many FIR responses at the 24-h post-test (mean of 93%).

A participant with a lower likelihood of assigning a common cause to incongruent auditory and visual speech perceives the McGurk effect less often (Fig. 4d–f). For this participant, the integrated audiovisual representation straddles the *ba* and *da* regions of representational space, leading to a mixture of both responses (due to noisy sensory encoding) when the McGurk stimulus is presented repeatedly. An error signal is produced by the discrepancy between the auditory and audiovisual representations, leading to a shift in the auditory representation (although the error signal and resulting shift is smaller than in a participant who always perceives the McGurk effect). In the auditory-only post-test, noisy encoding means that the encoded representation can fall in either the *ba* or *da* region of representational space, corresponding to an intermediate degree of FIR. This result models the group of participants shown in Fig. 2 who reported a moderate number of FIR responses at the 24-h post-test (mean of 57%).

A participant with a low likelihood of assigning a common cause to incongruent auditory and visual speech will almost never perceive the McGurk

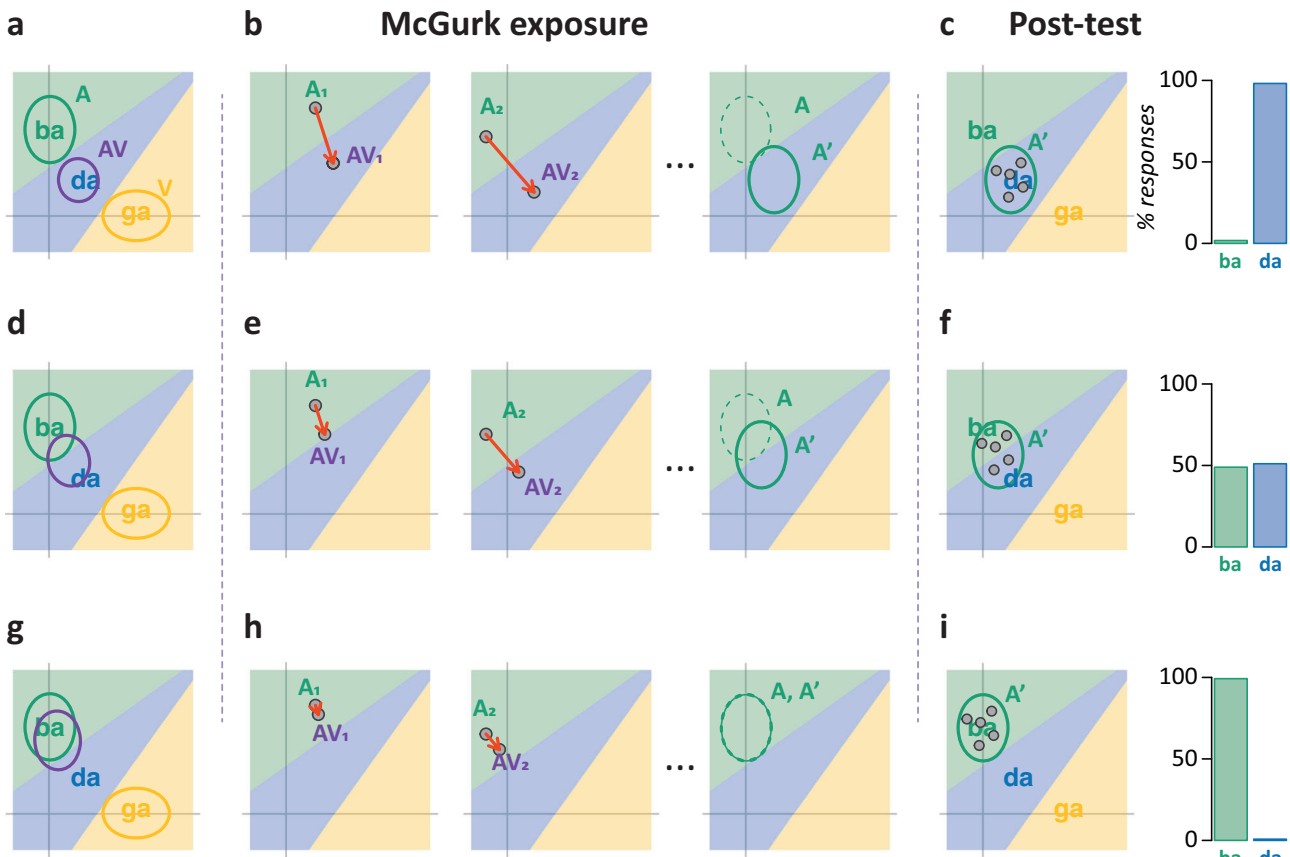

**Fig. 4 | Fusion-induced recalibration results from causal inference of multisensory speech. a** The top row illustrates the model fit for a participant who experiences strong FIR. The model posits an internal representational space for speech tokens; for simplicity, a two-dimensional space with three tokens is shown. During the presentation of a McGurk stimulus consisting of auditory *ba* paired with visual *ga*, three internal representations are created. The first representation is auditory (A; green ellipse). The second representation is visual (V; yellow ellipse). The third representation is audiovisual (AV; purple ellipse). The audiovisual representation is the average of the auditory and visual representations, weighted by the likelihood that they arise from a common cause (the same talker). For a participant who places a high likelihood on a common cause, the integrated representation is midway between the auditory and visual representations, in the *da* region of representational space. **b** During the presentation of McGurk stimuli during the exposure phase, sensory noise causes the encoded location of each representation (gray points) to fall at a different location somewhere within the 95% distribution ellipses shown in *A* (the encoded visual location, not shown, also varies). The integrated audiovisual representation always lies in the *da* region of representational space, leading to mainly McGurk fusion percepts. The difference between the encoded A and AV locations creates an error signal (red arrows). Two trials are shown (subscripts 1 and 2). Repeated errors induce a shift in the auditory representation from its original location (A, ellipse with green dashed line) to a new location that overlaps the integrated audiovisual representation (A', ellipse with solid green line), eliminating the error signal. **c** During the 24-h post-test, the shift in the auditory representation can be measured. The encoded auditory location varies from presentation to presentation due to sensory noise (each gray point represents the location in one presentation), but all fall within the A' ellipse, producing a

preponderance of FIR *da* percepts (bar graph). **d** The middle row illustrates the model fit for a participant who experiences moderate FIR, due to their tendency to estimate a moderate likelihood of a common cause for incongruent auditory and visual speech. For this type of participant, the integrated audiovisual representation is closer to the auditory representation, straddling the *ba* and *da* regions of representational space. **e** Sensory noise causes the encoded location to vary from trial to trial, resulting in a mixture of percepts across repeated presentations of the same McGurk stimulus. In the first trial, the encoded audiovisual representation is in the *ba* region of space, resulting in a *ba* percept, while in the second trial, the encoded audiovisual representation is in the *da* region of space. The difference between the encoded A and AV locations creates an error signal (red arrows) inducing a partial shift in the auditory representation from its original location (A) to a new location that straddles the *ba* and *da* boundary (A'). **f** In the 24-h post-test, sensory noise causes the encoded location to vary from presentation to presentation within the A' distribution, sometimes falling in the *ba* region of representational space and sometimes in the *da* region, producing a moderate number of FIR (*da*) percepts. **g** The bottom row illustrates the model fit for participants who do not experience FIR. Because they estimate a low likelihood of a common cause for incongruent auditory and visual speech, the integrated audiovisual representation is similar to the auditory representation. **h** Across repeated presentations of the McGurk stimulus, the integrated audiovisual representation always lies in the *ba* region of representational space, resulting in the absence of McGurk percepts and error signals. The auditory representation remains unchanged. **i** In the 24-h post-test, the encoded auditory location always falls in the *ba* region of representational space, resulting in no FIR (*da*) percepts.

effect (Fig. 4g–i). For this participant, the integrated audiovisual representation falls primarily in the *ba* region of representational space because of the high weight given to the auditory-only representation. This results in only a small error signal and little or no shift in the auditory representation. In the post-test, this means that the auditory stimulus is perceived veridically, corresponding to the absence of FIR. This result models the participants shown in Fig. 2 who never experienced the McGurk effect or FIR.

The CIMS model makes quantitative predictions about perception during the auditory-only post-test based on each participant's McGurk perception during the 14-day exposure period. The correspondence between the CIMS model prediction and the actual participant data was assessed by correlating the mean predicted FIR responses with each individual's actual percent FIR. The model accurately predicted participants' perception, $r_{28} = 0.75$, $p = 10^{-6}$ (95% CI: 0.5, 0.9).

## Discussion

Authors JFM and MSB repeatedly viewed AM's McGurk stimulus for teaching and research, leading to a surprising change in auditory perception that persists to the present day, many years later. This phenomenon, termed fusion-induced recalibration (FIR), was replicated in two groups of naive participants who underwent fourteen days of brief daily exposure to an audiovisual McGurk stimulus. Daily McGurk exposure altered the perception of an unambiguous auditory token so that it evoked the fusion percept: *baba* was perceived veridically in the pre-test but was perceived as *dada* in the post-test.

It is important to emphasize that FIR was found only in McGurk perceivers. Without the error signal introduced by the McGurk effect, the perception of unambiguous auditory syllables was stable over time. McGurk non-perceivers were exposed to the exact same stimuli as were McGurk perceivers, but unlike McGurk perceivers, they showed no changes in auditory perception.

In order to provide a quantitative explanation for FIR, we turned to the CIMS model of audiovisual speech perception[1,3]. The CIMS model incorporates Bayesian inference, with cues weighted by their reliability[2,21,22]. Bayesian integration is thought to underlie multisensory integration for continuous sensory variables such as spatial location as well as categorical variables such as speech[23].

The CIMS model was modified to incorporate an error signal[7-9,19,20]. The error signal reflects the difference between the audiovisual representation of the stimulus (the fusion percept of *dada*) and the auditory representation of the stimulus (*baba*). The error signal does not arise from external corrective feedback since no feedback was ever given in the experiment. Instead, the error signal is internal to the participant, the result of a conflict between separate audiovisual and auditory representations of speech content. The error signal prompts the auditory representation to shift towards the audiovisual representation with the result that on subsequent audiovisual trials, the auditory component of the McGurk stimulus becomes encoded as the fusion percept, eliminating the error signal. The change in auditory encoding (phonetic recalibration) was measured in auditory-only trials in the post-test.

In the CIMS model, the strength of the error signal is related to the effectiveness of the McGurk stimulus for a given participant. For participants who do not perceive the McGurk effect, there is no error signal as the auditory stimulus and the audiovisual percept are both *baba*. The absence of an error signal means there is no prompt to shift the auditory representation, resulting in the absence of recalibration in the auditory-only post-test. For participants who sometimes perceives the McGurk effect, there is a moderate error signal and shift in the auditory representation, resulting in an intermediate degree of recalibration in the post-test. For frequent McGurk perceivers, there is a strong error signal and a large shift in the auditory representing, resulting in a large degree of recalibration. Incorporating error-driven shifts in auditory representations allowed the CIMS model to successfully predict individual differences in FIR across participants.

A key point is that FIR was both talker-specific and syllable-specific. That is, not all auditory *ba* stimuli shifted to the *da* percept which would indeed be problematic for accurate speech perception. Instead, only the trained token was recalibrated. This is consistent with a large literature showing that participants are able to encode different auditory-to-phonetic mappings for different talkers[24-26].

### Limitations

There are a number of important questions about FIR that remain to be answered. Although the testing time was brief on each exposure day (5 repetitions, ~10 s), training extended over 14 days, a challenge to participant retention. It would be valuable to ascertain if FIR is inducible with fewer training days. A related question concerns the stimulus domain in which FIR is effective. Different McGurk stimuli vary in their perceptual efficacy[11,27] but the variability is lawful, so that efficacy is predictable from participant to participant[28]. Based on this observation, the main experiment examined a strong McGurk stimulus (AM's *baba/gaga*) and a weak one

(AN's *ba/ga*), finding that FIR was reliably induced only by the strong stimulus. Some *ba/ga* pairings evoke very few fusion percepts[29,30] and repeated exposure to these weak McGurk stimuli should not produce FIR.

The present study examined only the original McGurk syllable pairings of *ba/ga* and *pa/ka*. An important question for future research is whether FIR can be induced with other incongruent syllable pairings, such as auditory *ba* with visual *fa*[31-33]. In the present study, perceptual changes were induced and tested with the same token (e.g., AM's auditory *baba*). Would other utterances from AM, either syllables or entire words, also exhibit FIR? More broadly, how resistant is FIR to changes in the talker's voice, such as shifts in the fundamental frequency?

Another unanswered question is whether it is possible to unlearn FIR. In this scenario, a participant who experiences FIR for AM's auditory *baba* (perceiving *dada*) would be presented with AM's congruent audiovisual *baba* (presumably perceiving *baba* due to the additional visual speech information). This would give rise to an internal conflict in the participant between the auditory representation (*dada*) and the veridical audiovisual representation (*baba*). This error signal (in the opposite direction of that which originally induced FIR) might shift the auditory representation back to the *baba* region of representational space and eliminate FIR, even in participants who had maintained it for months or years.

A final unanswered question is the relationship between the short-duration (~seconds) recalibration observed in previous studies[7-9] and FIR's much longer timescale (months to years). One possible explanation is the 14-day exposure paradigm used to induce FIR. However, in previous studies, persistent recalibration was not observed despite the presentation of 256 audiovisual repetitions[34], many more than the 70 total repetitions in the present study; or the incorporation of a 24-h delay following audiovisual exposure[35]. A more likely explanation for the absence of persistent recalibration in previous studies is their stimulus set, which were often synthesized, ambiguous *ba/da* auditory speech tokens[7]; audiovisual stimuli that were relatively ineffective at evoking the McGurk effect; or both. In the CIMS model, an ambiguous *ba/da* auditory token would lie near the *ba/da* boundary in representational space, as would an ineffective McGurk stimulus. The proximity between the auditory and audiovisual representations would produce only a weak error signal and short-duration recalibration. In contrast, the large distance in representational space between AM's unambiguous auditory /ba/ and robust /da/ fusion percept produced a large error signal and long-lived recalibration.

### Relationship to the ventriloquism aftereffect

The original study of McGurk-induced changes in auditory perception was inspired by an illusion known as the ventriloquism aftereffect[7,9]. To study ventriloquism experimentally, simple audiovisual stimuli such as beeps and flashes are presented at the same time but in different locations. Observers estimate the position of the auditory stimulus as shifted toward the visual stimulus. The position of subsequent auditory-only stimuli is also perceived as shifted in the same direction (towards the previously presented visual stimulus). This phenomenon, known as the ventriloquism aftereffect (VAE), has been a rich source of information about multisensory integration, reviewed in[36,37]. Of particular relevance to FIR, studies of the VAE have shown that recalibration can occur rapidly, after only a single exposure to a discrepant audiovisual stimulus[38], and that recalibration can increase over time as training is repeated[39]. There have been no descriptions of VAE persisting for months or years, as we observed for FIR. This may be that unlike speech, in which perceivers maintain different acoustic-to-phonetic maps for different talkers, perceivers are more likely to use a common spatial framework for all auditory and visual stimuli.

### Other related perceptual phenomena: lexically-guided perception

Vroomen and colleagues[9,10,34,35,40] have pointed out parallels between FIR and another phenomenon that has come to be known as lexically-guided perception (LGP) in which the word context of an ambiguous sound alters perception[41]. One mystery discussed in[9] is that while FIR decayed over

seconds, LGP persists for minutes[42] or hours[43]. Our results clarify that FIR, if induced appropriately, can persist over timescales even longer than those described for LGP. A recent comparison between the benefits of visual speech and written text in understanding degraded auditory speech showed that visual speech provided a greater benefit than written speech, and this benefit persisted for more than a month, supporting the idea of a long-lasting influence of visual speech on auditory speech perception[44].

## Other related perceptual phenomena: verbal transformation effect

Repeated presentation of identical auditory or audiovisual speech tokens can induce perceptual changes, known as the verbal transformation effect[45,46]. Induction of the verbal transformation effect requires dozens of repetitions of an identical speech token, presented back-to-back without intervals between tokens. The speech token typically consists of a strong syllable and a weak syllable e.g., pa-*ta*, pa-*ta*, pa-*ta*; after repeated presentations with no gap, the syllables within the token reorganize e.g., *ta*-pa. This phenomenon is quite different than FIR. First, in the present study, identical tokens were never presented many times in succession. Instead, the stimuli of interest were randomly intermixed with control stimuli and the number of stimulus repetitions (five to ten on each exposure day) was many fewer than in verbal transformation studies (dozens or hundreds of back-to-back repetitions). In the present study, there was always a gap of several seconds between stimulus presentations, as participants were required to type their response into an open-choice response box before proceeding to the next stimulus presentation.

## Other related perceptual phenomena: selective adaptation

Repeated presentations of auditory speech tokens can induce selective adaptation[47–50]. The typical procedure for selective adaptation is to present a continuum of synthetic speech stimuli that vary across some acoustic parameter. The stimuli at either ends of the continuum are in different phonetic categories, while the middle stimuli are ambiguous (perceived as one category on some trials and as another category on other trials). Following repeated exposure to an identical speech sound, the canonical finding is that perception of ambiguous stimuli shifts away from the adapted stimulus, as if (putative) feature detectors for that stimulus were fatigued by the repetition.

Selective adaptation has also been examined in the context of audiovisual speech perception[31,51–55]. Dias and colleagues developed a clever variant of selective adaptation in which the test continuum was created by blurring the mouth region of the talker's face with progressively larger Gaussian filters[31]. Auditory /ba/ paired with unblurred visual /va/ produced the percept of /va/ in 94% of trials, but this percept became less frequent (with a concomitant increase in the perception of ba) as blurring increased. Following adaptation to visual va and audiovisual va, perception of the audiovisual continuum was modulated, but not following adaptation to auditory va, auditory ba, or visual ba, suggesting that auditory and visual speech information are not completely integrated at the level of selective adaptation.

There are profound differences between the circumstances in which selective adaptation and FIR are observed. The timescales of selective adaptation and FIR differ by orders of magnitude. Selective adaptation is observed for a few seconds following the repetition of the adapting stimulus; after a few seconds, adaptation dissipates, and additional top-up adapting stimuli must be presented. In contrast, FIR persists for days, weeks, or months, without any additional exposure to McGurk stimuli. Selective adaptation requires many (dozens to hundreds) presentations of an adapting stimulus in quick succession (typical interstimulus intervals of ~500 ms). In contrast, FIR was induced with 5 daily repetitions of a McGurk stimulus, with each repetition spaced by several seconds. In selective adaptation, a continuum of synthetic speech stimuli, including ambiguous speech, are presented; in contrast, in FIR, the auditory speech stimulus is a real talker speaking unambiguously.

Consistent with these major differences, no evidence for selective adaptation was observed in the present study. McGurk perceivers repeatedly experienced the fusion percept of *dada* on audiovisual exposure days. Under the adaptation account, this should decrease percepts of *dada* (as *dada* feature detectors grew fatigued). Instead, the opposite effect was observed, with McGurk perceivers reporting many more *dada* percepts in the auditory-only post-test than in the pre-test. Auditory perceivers repeatedly experienced the auditory percept of *baba* on audiovisual exposure days. If selective adaptation was at play, a decrease in percepts of *baba* would be expected, but instead, percepts of *baba* remained at 100% in the post-test. These results align with the absence of selective adaptation in short-term phonetic recalibration[8].

## Conclusions

Even very long-lasting perceptual phenomena, such as priming during picture naming, decays over the course of a year[56]. In contrast, in participants with high levels of FIR, there was no evidence of decay 360 days following the final audiovisual exposure. As a demonstration that the adult auditory system can have representations that are both malleable and long-lasting, FIR presents a fascinating neuroscience and modeling challenge[20]. FIR is just one example of the power of multisensory approaches to initiate plasticity in the nervous system[57–59].

Just as the McGurk effect has served as a useful tool for investigating audiovisual speech perception for almost a half-century, FIR may provide a useful probe of the neural and perceptual plasticity underlying important cognitive processes such as language learning during childhood and accent learning in adulthood[25,26]. The existence of FIR points to one of the most interesting properties of speech, in that it is both long-lasting (we can identify speech tokens in a language we have not heard in years or decades) and flexible enough to adapt to new talkers with different accents or speech mannerisms[24].

## Data availability

Data is available in the Dryad repository at https://doi.org/10.5061/dryad.4f4qrfjkw.

## Code availability

Code is available in the Zenodo repository at https://doi.org/10.5281/zenodo.10689633.

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

## Acknowledgements

This research was supported by NIH R01NS065395 and U01NS113339. The funders had no role in study design, data collection and analysis, decision to publish, or preparation of the manuscript.

## Author contributions

John Magnotti designed the experiments, analyzed the data, and prepared the manuscript and figures. Anastasia Lado designed the experiments, collected the data, and analyzed the data. Yue Zhang designed the experiments, collected the data, and analyzed the data. Arnt Maasø created the stimuli. Audrey Nath created the stimuli. Michael Beauchamp designed the experiments, analyzed the data, and prepared the manuscript and figures.

## Competing interests

The authors declare no competing interests.
