## [Peer Review File · Communications Psychology]

13th Sep 23

Dear Professor Beauchamp,

Thank you for your patience during the peer-review process. Your manuscript titled "Fusion-Induced Recalibration: A long-lasting aftereffect of the McGurk effect" has now been seen by 3 reviewers, and I include their comments at the end of this message. They find your work of interest, but raised some important points. We are interested in the possibility of publishing your study in *Communications Psychology*, but would like to consider your responses to these concerns and assess a revised manuscript before we make a final decision on publication.

We therefore invite you to revise and resubmit your manuscript, along with a point-by-point response to the reviewers. Please highlight all changes in the manuscript text file.

Editorially, we ask you to pay particular attention to the following:

The reviewers highlight a few methodological issues that require additional analyses. This includes the concerns regarding the split of the participants into a fusion and non-fusion group as well as the use of the non-fusion group as a statistical control (see Reviewer #2). Please also address all concerns regarding the CIMS model and clearly state the error signals and the assumptions for the internal signals in the CIMS model.

Reviewer #1 highlights that while there is interest in the phenomenological observation offered by your study, there are a range of important empirical questions that remain unaddressed. We very strongly recommend you include a conceptual replication of the present finding that allows you to test the conditions for and generalization of the effect. We appreciate that a full replication of the 1yr follow-up is beyond the scope of a revision.

Please also address other methodological concerns, such as how accurate perceptual presentation was ensured.

Finally, while the motivation for the study may have been an anecdotal observation (and this motivation should be stated faithfully), we ask that you constructively engage with all relevant literature, as suggested by the reviewers, including literature on other forms of audiovisual recalibration.

Please use the following link to submit your revised manuscript, point-by-point response to the referees' comments (which should be in a separate document to any cover letter) and the completed checklist:

[link redacted]

Please do not hesitate to contact me if you have any questions or would like to discuss these revisions further. We look forward to seeing the revised manuscript and thank you for the opportunity to review your work.

Best regards,

Jixing Li

Jixing Li, PhD
Editorial Board Member
Communications Psychology
orcid.org/0000-0002-5210-6224

EDITORIAL POLICIES AND FORMATTING

Editorial Policy: Policy requirements (Download the link to your computer as a PDF.)

Furthermore, please align your manuscript with our format requirements, which are summarized on the following checklist:

Communications Psychology formatting checklist

and also in our style and formatting guide Communications Psychology formatting guide .

* TRANSPARENT PEER REVIEW: Communications Psychology uses a transparent peer review system. This means that we publish the editorial decision letters including Reviewers' comments to the authors and the author rebuttal letters online as a supplementary peer review file. However, on author request, confidential information and data can be removed from the published reviewer reports and rebuttal letters prior to publication. If your manuscript has been previously reviewed at another journal, those Reviewers' comments would not form part of the published peer review file.

* CODE AVAILABILITY: All Communications Psychology manuscripts must include a section titled

"Code Availability" at the end of the methods section. In the event of publication, we require that the custom analysis code supporting your conclusions is made available in a publicly accessible repository; at publication, we ask you to choose a repository that provides a DOI for the code; the link to the repository and the DOI will need to be included in the Code Availability statement. Publication as Supplementary Information will not suffice. We ask you to prepare code at this stage, to avoid delays later on in the process.

* DATA AVAILABILITY:

All Communications Psychology manuscripts must include a section titled "Data Availability" at the end of the Methods section or main text (if no Methods). More information on this policy, is available at <http://www.nature.com/authors/policies/data/data-availability-statements-data-citations.pdf>.

At a minimum the Data availability statement must explain how the data can be obtained and whether there are any restrictions on data sharing. Communications Psychology strongly endorses open sharing of data. If you do make your data openly available, please include in the statement:

We recommend submitting the data to discipline-specific, community-recognized repositories, where possible and a list of recommended repositories is provided at <http://www.nature.com/sdata/policies/repositories>.

If a community resource is unavailable, data can be submitted to generalist repositories such as [figshare](http://www.figshare.com) or [Dryad Digital Repository](http://www.dryad.org). Please provide a unique identifier for the data (for example a DOI or a permanent URL) in the data availability statement, if possible. If the repository does not provide identifiers, we encourage authors to supply the search terms that will return the data. For data that have been obtained from publicly available sources, please provide a URL and the specific data product name in the data availability statement. Data with a DOI should be further cited in the methods reference section.

REVIEWERS' EXPERTISE:

Reviewer #1 multisensory processing
Reviewer #2 audiovisual integration, learning
Reviewer #3 CIMS model

REVIEWERS' COMMENTS:

Reviewer #1 (Remarks to the Author):

Summary

In this short article, authors Magnotti and Beauchamp of the present manuscript report that

repeated exposure to a McGurk stimulus of a specific speaker (A.M.) resulted in long-lasting changes to their auditory percepts of this speaker, as both the auditory-only presentation and audiovisual presentation which included the face elicit the illusory McGurk fusion percept. Their subjective experience of the effect lasted years after exposure.

To investigate whether repeated exposure produces similar effects in naïve observers, and to use their causal inference model to account for the effect, they conducted a single experiment. In a pre-test phase, 30 participants reported their perception of A.M.'s "baba" stimulus, among other randomly intermixed auditory stimuli. Almost everyone correctly perceived "baba" to begin. After the pre-test, participants were exposed to A.M.'s McGurk stimulus (auditory baba, visual gaga) 5 times a day for 14 days. Participants that always perceived the auditory component of the stimulus stayed that way following exposure. However, 10/16 participants that perceived the fused McGurk percept now perceived the auditory baba stimulus as "dada." (The remaining 6/16 participants still perceived the baba stimulus veridically). Interestingly, the auditory-only recalibration was speaker-specific, as "baba" perception was unaltered in two control stimuli by male talkers.

A causal inference model accounted for this effect via the auditory representation shifting towards the visual representation, similar to other audiovisual recalibration effects (e.g., the ventriloquist aftereffect in audiovisual spatial perception).

I find little to critique about the present study: the statistical methods are sound, and the finding is of interest to the field. However, the study's brevity opens up many unknown questions, which I will outline below. I leave it up to the editor if these should be addressed in a revision or by future work.

Major Concerns

- Can the authors comment on how this phenomenon relates to the literature on another form of audiovisual recalibration: the ventriloquist aftereffect? For example, Wozny and Shams 2011 showed that repeated presentation of spatially misaligned audiovisual stimuli resulted in a spatial shift in the perception of subsequent auditory stimuli. This finding was shown to be best accounted for by shifts in the representations (likelihoods) of auditory space, similar to what the authors show here. I think that the VAE literature and the Wozny and Shams paper should be discussed in the final sections of the paper, because I think there are some points of congruence between that paper and this one. Please also cite this prior work of Vikranth Bejjanki, which predates the CIMS model: <https://journals.plos.org/plosone/article?id=10.1371/journal.pone.0019812>
- There are a number of questions that this study leaves unaddressed, including:
 - o How systematically manipulating exposure duration changes the effect (i.e., we have no idea about the relationship between exposure time and how long-lasting the effect is)
 - o Whether repeated exposure to McGurk stimuli not only shifts the representations of auditory stimuli, but also the prior probability of inferring a common cause (i.e., the likelihood of integrating . . . similar to Odegaard et al. 2017, PeerJ)
 - o Whether this effect occurs for stimuli other than auditory baba/ visual gaga
 - o Whether this effect is specific to the speaker that was used for the stimulus (A.M.)
 - o If there are ways to DECREASE the effect (e.g., by repeatedly presenting "baba" auditory stimuli with visual faces that DON'T result in a McGurk percept, so that instead of repeated integration, participants experience repeated segregation).

Reviewer #2 (Remarks to the Author):

Summary

This manuscript reports the results of a study employing the McGurk effect to induce long-term perceptual changes to auditory stimuli that are substantiated by a modified perceptual model based on causal inference.

General Comments

This study may add to a large corpus of studies in phonetic recalibration, changes in auditory perception based on paired visual stimuli. However, there are several problems with the manuscript that need to be addressed.

1. The motivation for the study seems to be heavily dependent on the anecdotal experiences of the authors with very little attention given to the broad empirical and theoretical literature concerning visual speech influences on auditory phonetic recalibration.
2. The stark dichotomy between participants who perceived a fusion stimulus and those who did not is striking, and highly unusual for the McGurk combination used, especially when participants were allowed to give open-ended responses. I encourage the authors to closely inspect the data for some kind of explanation for this dichotomy. For example, were these groups trained at different times? Was there a change in the auditory intensity of the stimuli? Were participants instructed to have their computers set to a comfortable listening level?
3. Using the non-fusion group as a statistical control is problematic. There was no theory-driven expectation for such a group to exist and no motivation provided for using them as the comparison group. All participants' performances should be compared to their own baseline and training data to measure changes in perception.
4. Finding visual-induced changes in the perception of "clear" auditory stimuli (/baba/ was identified as "baba" ~94% of the time) is irregular. Visual-induced phonetic recalibration is typically observed when the auditory component is ambiguous (identified as "baba" ~50% of the time) (e.g., Bertelson et al., 2003; Lüttke et al., 2016; Vroomen & Baart, 2012). There needs to be some motivation for an expected recalibration effect using a clear auditory component and an explanation for the observed recalibration given the previous literature.
5. Greater explanation is needed for the CIMS model. In particular, there must be some theoretical motivation and explanation of the mathematical parameters of the model to understand the underlying mechanisms the model tests in the processing of auditory and visual stimuli to induce the recalibration observed in the human data.

References

- Bertelson, P., Vroomen, J., & de Gelder, B. (2003). Visual recalibration of auditory speech identification: A McGurk aftereffect. *Psychological Science*, 14(6), 592-597.
- Lüttke, C. S., Ekman, M., van Gerven, M. A. J., & de Lange, F. P. (2016). McGurk illusion recalibrates subsequent auditory perception. *Scientific Reports*, 6, 32891. <https://doi.org/10.1038/srep32891>

Vroomen, J., & Baart, M. (2012). Phonetic recalibration in audiovisual speech. In M. M. Murray & M. T. Wallace (Eds.), *The Neural Bases of Multisensory Processes* (pp. 363-379). CRC Press.

Reviewer #3 (Remarks to the Author):

In this paper Magnotti et al continue some of their previous work on the McGurk effect by examining the long term learning effects after exposure during a two week period. The authors find that the effect can be long lasting (up to a year), and that a Bayesian causal inference model can explain the learning endpoint well.

I think the article is appropriate for this journal, although I will mostly let others decide that.

Experimental methods seem appropriate, although it would be good to know if anything was done to ensure that participants had a good enough perceptual presentation (screen size, headphones, etc.)

Authors should make a note that the proposed error signal, is different from a typical external error signal, here it is the difference between a unimodal audio signal and a mean posterior after causal inference. This is not unreasonable to learn based on an internal error signal, but should be clearly stated.

Regarding this internal signal, there is an assumption that the learning/adaptation endpoint is given by the mean of the posterior mixture model, that is the out put from the causal inference model. This might not necessarily be the case, as other studies have shown tendencies not to shift completely. This assumption should be made clearer, as it currently is just one sentence in the Methods.

The authors show that the correlation between the predicted and empirical learning endpoint are highly correlated, but a correlation could still hide differences across the distributions. As a minimum the correlation should be shown (ie a scatter plot).

And please put some type of error bars on figure 2C-right, just to show the variation.

Response to Referees (Rebuttal letter)

We thank the three Referees for their careful reading of the manuscript. The reviewers raised important concerns about our initial submission which we have addressed fully in the resubmission. Importantly, we have collected additional data from a new group of participants. As shown in the new Figure 3, we replicated our finding of fusion-induced recalibration (FIR) in the original stimulus from talker A.M. as well as another McGurk stimulus from a different talker (A.N.) composed of a different syllable pairing (auditory *pa* with visual *ka*). Comments and responses are numbered in the format *Qx.y/Ax.y* for ease of reference.

Referee #1

Summary

In this short article, authors Magnotti and Beauchamp of the present manuscript report that repeated exposure to a McGurk stimulus of a specific speaker (A.M.) resulted in long-lasting changes to their auditory percepts of this speaker, as both the auditory-only presentation and audiovisual presentation which included the face elicit the illusory McGurk fusion percept. Their subjective experience of the effect lasted years after exposure.

To investigate whether repeated exposure produces similar effects in naïve observers, and to use their causal inference model to account for the effect, they conducted a single experiment. In a pre-test phase, 30 participants reported their perception of A.M.'s "baba" stimulus, among other randomly intermixed auditory stimuli. Almost everyone correctly perceived "baba" to begin. After the pre-test, participants were exposed to A.M.'s McGurk stimulus (auditory baba, visual gaga) 5 times a day for 14 days. Participants that always perceived the auditory component of the stimulus stayed that way following exposure. However, 10/16 participants that perceived the fused McGurk percept now perceived the auditory baba stimulus as "dada." (The remaining 6/16 participants still perceived the baba stimulus veridically). Interestingly, the auditory-only recalibration was speaker-specific, as "baba" perception was unaltered in two control stimuli by male talkers.

A causal inference model accounted for this effect via the auditory representation shifting towards the visual representation, similar to other audiovisual recalibration effects (e.g., the ventriloquist aftereffect in audiovisual spatial perception).

I find little to critique about the present study: the statistical methods are sound, and the finding is of interest to the field. However, the study's brevity opens up many unknown questions, which I will outline below. I leave it up to the editor if these should be addressed in a revision or by future work.

Major Concerns

Q1.1 Can the authors comment on how this phenomenon relates to the literature on another form of audiovisual recalibration: the ventriloquist aftereffect? For example, Wozny and Shams 2011 showed that repeated presentation of spatially misaligned audiovisual stimuli resulted in a spatial shift in the perception of subsequent auditory stimuli. This finding was shown to be best accounted for by shifts in the representations (likelihoods) of auditory space, similar to what

the authors show here. I think that the VAE literature and the Wozny and Shams paper should be discussed in the final sections of the paper, because I think there are some points of congruence between that paper and this one. Please also cite this prior work of Vikranth Bejjanki, which predates the CIMS model: <https://journals.plos.org/plosone/article?id=10.1371/journal.pone.0019812>

A1.1 We agree that the ventriloquism aftereffect is very relevant and now write in the discussion:

Relationship to the ventriloquism aftereffect

The original study of McGurk-induced changes in auditory perception was inspired by an illusion known as the ventriloquism aftereffect (Bertelson et al., 2003; Vroomen and Baart, 2012). To study ventriloquism experimentally, simple audiovisual stimuli such as beeps and flashes are presented at the same time but in different locations. Observers estimate the position of the auditory stimulus as shifted toward the visual stimulus. The position of subsequent auditory-only stimuli are also perceived as shifted in the same direction (towards the previously presented visual stimulus). This phenomenon, known as the ventriloquism aftereffect (VAE), has been a rich source of information about multisensory integration, reviewed in (Bruns, 2019; Chen and Vroomen, 2013). Of particular relevance to FIR, studies of the VAE have shown that recalibration can occur rapidly, after only a single exposure to a discrepant audiovisual stimulus (Wozny and Shams, 2011) and that recalibration can increase over time as training is repeated (Bruns and Röder, 2019). There have been no descriptions of VAE persisting for months or years as we observed for FIR. This may be that unlike for speech, in which perceivers maintain different acoustic-to-phonetic maps for different talkers, perceivers are more likely to use a common spatial framework for all auditory and visual stimuli.

We agree that the Bejjanki work is very relevant, and now write in the discussion:

Bayesian integration is thought to underlie multisensory integration for continuous sensory variables such as spatial location as well as categorical variables such as speech (Bejjanki et al., 2011).

Q1.2 There are a number of questions that this study leaves unaddressed.

A1.2 We have performed a replication study in a new group of participants to help answer the reviewers' questions. Of course, as with any new phenomenon, a complete understanding of FIR will require many additional studies.

Q1.3 How systematically manipulating exposure duration changes the effect (i.e., we have no idea about the relationship between exposure time and how long-lasting the effect is)

A1.3 We agree that exposure duration will be an important variable to manipulate in future studies. Determining the relationship between McGurk exposure and FIR duration will require a large, long-term study as FIR can last for weeks, months or years.

Q1.4 Whether repeated exposure to McGurk stimuli not only shifts the representations of auditory stimuli, but also the prior probability of inferring a common cause (i.e., the likelihood of integrating . . . similar to Odegaard et al. 2017, PeerJ)

A1.4 This is an interesting question. In the first Experiment, we presented two McGurk stimuli (S1 and S2) across 14 days but did not see any obvious increase in McGurk responses within subject for either stimulus across days, suggesting that the prior probability for inferring a common cause was not changing following repeated McGurk exposure. A stronger test would be to show the same untrained McGurk stimuli only in the pre-test and post-test and compare the number of McGurk responses before and after, with the expectation of a significant difference if the prior probability of $C = 1$ was shifted. However, it is important to point out that any change in the prior probability of inferring a common cause for *audiovisual* stimuli cannot explain the changed perception (FIR) we observe for *auditory-only* stimuli in the 24-hour post-test.

Q1.5 Whether this effect occurs for stimuli other than auditory baba/ visual gaga

A1.5 In the replication study, we now present the second pairing (described by McGurk and MacDonald in their original description of the effect) of auditory *pa* paired with visual *ka*, resulting in the illusory fusion percept of *ta*. We show that FIR is also observed for this pairing.

Q1.6 Whether this effect is specific to the speaker that was used for the stimulus (A.M.)

A1.6 In the replication study, we show that FIR could also be induced with a McGurk stimulus recorded by a different talker (author A.N.). FIR was specific for the McGurk stimulus and for the talker; participants could experience FIR with A.M.'s McGurk stimulus but not A.N.'s McGurk stimulus, or *vice-versa*. The perception of control stimuli consisting of the same syllables from different talkers was always veridical, indicating that FIR is highly specific.

Q1.7 If there are ways to DECREASE the effect (e.g., by repeatedly presenting “baba” auditory stimuli with visual faces that DON’T result in a McGurk percept, so that instead of repeated integration, participants experience repeated segregation).

A1.7 We agree that additional experiments aimed at decreasing FIR (or the McGurk effect) with various manipulations would be very interesting to conduct.

Referee #2 (Remarks to the Author):

Summary

This manuscript reports the results of a study employing the McGurk effect to induce long-term perceptual changes to auditory stimuli that are substantiated by a modified perceptual model based on causal inference.

General Comments

This study may add to a large corpus of studies in phonetic recalibration, changes in auditory perception based on paired visual stimuli. However, there are several problems with the manuscript that need to be addressed.

Q2.1 The motivation for the study seems to be heavily dependent on the anecdotal experiences of the authors with very little attention given to the broad empirical and theoretical literature concerning visual speech influences on auditory phonetic recalibration.

A2.1 We have added additional *Discussion* on related phenomena, such as the ventriloquism aftereffect.

Q2.2 The stark dichotomy between participants who perceived a fusion stimulus and those who did not is striking, and highly unusual for the McGurk combination used, especially when participants were allowed to give open-ended responses. I encourage the authors to closely inspect the data for some kind of explanation for this dichotomy. For example, were these groups trained at different times? Was there a change in the auditory intensity of the stimuli? Were participants instructed to have their computers set to a comfortable listening level?

A2.2 We agree that the dichotomy between McGurk fusion perceivers and non-perceivers is very interesting and have investigated it extensively in previous studies. For instance, in Basu Mallick *et al.* (Basu Mallick *et al.*, 2015), 165 undergraduates were tested in-person. For any given McGurk stimulus, there was a sharp dichotomy between fusion perceivers and non-perceivers. The plot at right shows the number of McGurk responses (percent fusion) in every participant, ordered from smallest to largest for a single stimulus (stimulus #9). Rather than a normal distribution, most participants reported either no fusion responses (0%) or all fusion responses (100%). In the Basu Mallick study, this dichotomy was observed for every stimulus tested, including the A.M. stimulus used in the present study.

In the Basu Mallick study, all participants were tested with the exact same experimental setup. This means that these individual differences cannot be attributed to changes in the auditory intensity of the stimuli or differences in the computers used for testing. While the causes of these individual differences in McGurk perception is an area of active investigation, one source of variation seems to be the pattern of eye movements made by individuals when they view faces (Gurler *et al.*, 2015; Rennig *et al.*, 2020).

In previous studies, we have compared on-line and in-person tests of the McGurk effect and found very similar results, as in Figure 3 from (Magnotti *et al.*, 2018)

Figure 3. No difference between in-person and online testing. (A) Accuracy of responses for auditory-only syllables, mean and standard error within in-person (orange) and online (red) participants, p -value from t -test. (B) Percent of McGurk responses averaged within in-person and online participants. (C) Individual participant McGurk prevalence for in-person participants. One symbol per participant, sorted by magnitude, error bars show standard error across stimuli. (D) Individual participant McGurk prevalence for online participants.

Q2.3 Using the non-fusion group as a statistical control is problematic. There was no theory-driven expectation for such a group to exist and no motivation provided for using them as the comparison group. All participants' performances should be compared to their own baseline and training data to measure changes in perception.

A2.3 We have revised the methods to make it clear that we did not use the non-fusion group as a statistical control. All analyses were done within participants using mixed-effects models, the statistical best practice.

Q2.4 4. Finding visual-induced changes in the perception of “clear” auditory stimuli (/baba/ was identified as “baba” ~94% of the time) is irregular. Visual-induced phonetic recalibration is typically observed when the auditory component is ambiguous (identified as “baba” ~50% of the time) (e.g., Bertelson et al., 2003; Lüttke et al., 2016; Vroomen & Baart, 2012). There needs to be some motivation for an expected recalibration effect using a clear auditory component and an explanation for the observed recalibration given the previous literature.

A2.4 Yes, it is extremely irregular! In the revision we include a replication with a completely new set of participants to confirm the existence of this interesting and previously undescribed phenomenon. The body of the manuscript is devoted to experiments and models that bring us closer to a better understanding of FIR.

Q2.5 Greater explanation is needed for the CIMS model. In particular, there must be some theoretical motivation and explanation of the mathematical parameters of the model to understand the underlying mechanisms the model tests in the processing of auditory and visual stimuli to induce the recalibration observed in the human data.

A2.5 We now give more examples in Figure 2 of how the CIMS model fits data from different FIR participants; our previous publications on the CIMS model provide additional details on the theoretical motivation and explanations of the mathematical parameters (Magnotti et al., 2020; Magnotti and Beauchamp, 2017).

Referee #3 (Remarks to the Author):

In this paper Magnotti et al continue some of their previous work on the McGurk effect by examining the long term learning effects after exposure during a two week period. The authors find that the effect can be long lasting (up to a year), and that a Bayesian causal inference model can explain the learning endpoint well.

I think the article is appropriate for this journal, although I will mostly let others decide that.

Q3.1 Experimental methods seem appropriate, although it would be good to know if anything was done to ensure that participants had a good enough perceptual presentation (screen size, headphones, etc.)

A3.1 In previous studies, we have compared on-line and in-person tests of the McGurk effect and found very similar results, as in Figure 3 from (Magnotti et al., 2018)

Figure 3. No difference between in-person and online testing. (A) Accuracy of responses for auditory-only syllables, mean and standard error within in-person (orange) and online (red) participants, p -value from t -test. (B) Percent of McGurk responses averaged within in-person and online participants. (C) Individual participant McGurk prevalence for in-person participants. One symbol per participant, sorted by magnitude, error bars show standard error across stimuli. (D) Individual participant McGurk prevalence for online participants.

Clearly, the McGurk effect is effective even under the variety of viewing and listening conditions experienced by different online participants. High variability is observed even when all participants are tested in-person with an identical audiovisual presentation system (yellow data in Figure 3 above). In the current experiment, if participants did not experience the McGurk effect because of their viewing conditions, then they were also not able to demonstrate recalibration. In this case, our data would be an underestimate of the true prevalence of FIR.

Q3.2 Authors should make a note that the proposed error signal, is different from a typical external error signal, here it is the difference between a unimodal audio signal and a mean posterior after causal inference. This is not unreasonable to learn based on an internal error signal, but should be clearly stated. Regarding this internal signal, there is an assumption that the learning/adaptation endpoint is given by the mean of the posterior mixture model, that is the output from the causal inference model. This might not necessarily be the case, as other studies have shown tendencies not to shift completely. This assumption should be made clearer, as it currently is just one sentence in the Methods.

A3.2 This is a key point and we now write in the manuscript:

The error signal reflects the difference between the audiovisual representation of the stimulus (the fusion percept of /dada/) and the auditory representation of the stimulus (/baba/). The error signal does not arise from external corrective feedback since no feedback was ever given in the experiment. Instead, the error signal is internal to the participant, the result of the conflict between the audiovisual and auditory representations. The error signal prompts the auditory representation to shift towards the audiovisual representation with the result that on subsequent

audiovisual trials, the auditory component of the McGurk stimulus becomes encoded as the fusion percept, eliminating the error signal. The change in auditory encoding (phonetic recalibration) was measured in auditory-only trials in the post-test.

Q3.3 The authors show that the correlation between the predicted and empirical learning endpoint are highly correlated, but a correlation could still hide differences across the distributions. As a minimum the correlation should be shown (ie a scatter plot). And please put some type of error bars on figure 2C-right, just to show the variation.

A3.3 We have revamped Figure 2 so that it better illustrates how the CIMS model explains the correlation between McGurk perception and FIR for three representative participants, while eliminating Figure 2C. The biggest driver of the model fit is that high McGurk predicts high FIR, so the scatter plot is uninformative, with many points at (0,0) and (1,1).

Response to Reviewers References

(see main manuscript for complete reference list)

Basu Mallick, D., F. Magnotti, J., S. Beauchamp, M., 2015. Variability and stability in the McGurk effect: contributions of participants, stimuli, time, and response type. *Psychon Bull Rev* 22, 1299–1307. <https://doi.org/10.3758/s13423-015-0817-4>

Bejjanki, V.R., Clayards, M., Knill, D.C., Aslin, R.N., 2011. Cue integration in categorical tasks: insights from audio-visual speech perception. *PLoS One* 6, e19812. <https://doi.org/10.1371/journal.pone.0019812>

Bertelson, P., Vroomen, J., De Gelder, B., 2003. Visual recalibration of auditory speech identification: a McGurk aftereffect. *Psychol Sci* 14, 592–7.

Bruns, P., 2019. The Ventriloquist Illusion as a Tool to Study Multisensory Processing: An Update. *Front Integr Neurosci* 13, 51. <https://doi.org/10.3389/fnint.2019.00051>

Bruns, P., Röder, B., 2019. Repeated but not incremental training enhances cross-modal recalibration. *Journal of Experimental Psychology: Human Perception and Performance* 45, 435–440. <https://doi.org/10.1037/xhp0000642>

Chen, L., Vroomen, J., 2013. Intersensory binding across space and time: A tutorial review. *Atten Percept Psychophys* 75, 790–811. <https://doi.org/10.3758/s13414-013-0475-4>

Gurler, D., Doyle, N., Walker, E., Magnotti, J., Beauchamp, M., 2015. A link between individual differences in multisensory speech perception and eye movements. *Atten Percept Psychophys*. <https://doi.org/10.3758/s13414-014-0821-1>

Magnotti, J.F., Beauchamp, M.S., 2017. A Causal Inference Model Explains Perception of the McGurk Effect and Other Incongruent Audiovisual Speech. *PLoS Comput Biol* 13, e1005229. <https://doi.org/10.1371/journal.pcbi.1005229>

Magnotti, J.F., Dzeda, K.B., Wegner-Clemens, K., Rennig, J., Beauchamp, M.S., 2020. Weak observer-level correlation and strong stimulus-level correlation between the McGurk effect and audiovisual speech-in-noise: A causal inference explanation. *Cortex* 133, 371–383. <https://doi.org/10.1016/j.cortex.2020.10.002>

Magnotti, J.F., Smith, K.B., Salinas, M., Mays, J., Zhu, L.L., Beauchamp, M.S., 2018. A causal inference explanation for enhancement of multisensory integration by co-articulation. *Sci Rep* 8, 18032. <https://doi.org/10.1038/s41598-018-36772-8>

Rennig, J., Wegner-Clemens, K., Beauchamp, M.S., 2020. Face viewing behavior predicts multisensory gain during speech perception. *Psychon Bull Rev* 27, 70–77. <https://doi.org/10.3758/s13423-019-01665-y>

Vroomen, J., Baart, M., 2012. Phonetic Recalibration in Audiovisual Speech, in: Murray, M.M., Wallace, M.T. (Eds.), *The Neural Bases of Multisensory Processes*, *Frontiers in Neuroscience*. CRC Press/Taylor & Francis, Boca Raton (FL).

Wozny, D.R., Shams, L., 2011. Recalibration of auditory space following milliseconds of cross-modal discrepancy. *The Journal of Neuroscience* 31, 4607–4612.
<https://doi.org/10.1523/JNEUROSCI.6079-10.2011>

9th Jan 24

Dear Professor Beauchamp,

Thank you for your patience during the peer-review process. Your manuscript titled "Fusion-Induced Recalibration: A long-lasting aftereffect of the McGurk effect" has now been seen by 3 reviewers, whose comments are appended below. You will see that while two reviewers are satisfied with your revisions, Reviewer #2 raised some additional substantial concerns that must be addressed before we can proceed with your manuscript.

Unless the existing data allows you to address the issue raised by the reviewer beyond doubt, you will be required to collect new data. We understand that this is a significant undertaking and are happy to set an extended deadline for the resubmission of 9 months or longer. If you cannot fully address the issue based on the existing data or by providing novel empirical evidence, we regret that you are best advised to seek publication elsewhere. In this case, please contact us so that we can close your file.

If you choose to revise the manuscript in response to Reviewer #2's concerns, please highlight all changes in the manuscript text file, and provide a detailed point-by-point reply to the reviewers.

If the revision process takes significantly longer than nine months, we will be happy to reconsider your paper at a later date, provided it still presents a significant contribution to the literature at that stage.

Please use the following link to submit your revised manuscript, point-by-point response to the Reviewers' comments with a list of your changes to the manuscript text (which should be in a separate document to any cover letter) and any completed checklist:

[link redacted]

Please do not hesitate to contact me if you have any questions or would like to discuss the required revisions further. Thank you for the opportunity to review your work.

Best regards,

Dr Antonia Eisenkoeck, Senior Editor

on behalf of

Jixing Li, PhD
Editorial Board Member
Communications Psychology
orcid.org/0000-0002-5210-6224

EDITORIAL POLICIES AND FORMATTING

Editorial Policy: Policy requirements (Download the link to your computer as a PDF.)

Furthermore, please align your manuscript with our format requirements, which are summarized on the following checklist:

Communications Psychology formatting checklist

and also in our style and formatting guide Communications Psychology formatting guide .

* **CODE AVAILABILITY:** All Communications Psychology manuscripts must include a section titled "Code Availability" at the end of the methods section. In the event of publication, we require that the custom analysis code supporting your conclusions is made available in a publicly accessible repository; please choose a repository that provides a DOI for the code; the link to the repository and the DOI must be included in the Code Availability statement. Publication as Supplementary Information will not suffice. We ask you to prepare and upload code at this stage, to avoid delays later on in the process.

* **DATA AVAILABILITY:**

All Communications Psychology research manuscripts must include a section titled "Data Availability" at the end of the Methods section or main text (if no Methods). More information on this policy, is available at <http://www.nature.com/authors/policies/data/data-availability-statements-data-citations.pdf>.

At a minimum the Data availability statement must explain how the data can be obtained and whether there are any restrictions on data sharing. Communications Psychology strongly endorses open sharing of data. If you do make your data openly available, please include in the statement:
- Unique identifiers (such as DOIs and hyperlinks for datasets in public repositories)

- Accession codes where appropriate
- If applicable, a statement regarding data available with restrictions
- If a dataset has a Digital Object Identifier (DOI) as its unique identifier, we strongly encourage including this in the Reference list and citing the dataset in the Data Availability Statement.

We recommend submitting the data to discipline-specific, community-recognized repositories, where possible and a list of recommended repositories is provided at <http://www.nature.com/sdata/policies/repositories>.

If a community resource is unavailable, data can be submitted to generalist repositories such as figshare or Dryad Digital Repository. Please provide a unique identifier for the data (for example a DOI or a permanent URL) in the data availability statement, if possible. If the repository does not provide identifiers, we encourage authors to supply the search terms that will return the data. For data that have been obtained from publicly available sources, please provide a URL and the specific data product name in the data availability statement. Data with a DOI should be further cited in the methods reference section.

Reviewer #1 (Remarks to the Author):

I am satisfied with the authors' replies to my comments on the first manuscript. I think this is an interesting behavioral finding that contributes to a growing literature on multisensory recalibration. Congrats on a nice paper.

Reviewer #2 (Remarks to the Author):

General Comments

I greatly appreciate the authors' efforts to address the concerns raised by myself and the other reviewers in this revised manuscript. The expanded discussion and the addition of another experiment give greater theoretical grounding for the results and specificity to the findings. I find the manuscript to be greatly improved.

I am, however, now greatly concerned that the results may be the result of something other than perceptual recalibration. A complimentary phenomenon to speech perception recalibration is the perceptual aftereffect known as "selective adaptation" (e.g., Cooper, 1974; Cooper & Lauritsen, 1974). In this phenomenon, perception of speech can be modulated by repeated presentation of within-modality speech stimuli. For example, perception of auditory speech stimuli along a /ba/-to-/va/ continuum can be modulated toward /ba/ (more stimuli along the continue perceived as "va") following repeated presentation of an auditory /ba/ stimulus. Similarly, perception of auditory stimuli along the same /ba/-to-/va/ continuum can be modulated toward /va/ (more stimuli along the continuum perceived as "ba") following repeated presentation of an auditory "va" stimulus.

Selective adaptation has famously been found to be strongest within-modality with very weak (if any) cross-sensory effects (Roberts & Summerfield, 1981) (see also Dias, 2016; Dias et al., 2016). What is observed in the studies reported in the current manuscript is described by the authors as a “recalibration” of the auditory stimuli toward the McGurk /da/ stimulus (auditory /ba/ toward perceived /da/). However, the results could also be described as a modulation of perception of the auditory /ba/ stimuli as less /ba/-like following “selective adaptation” to the auditory component of the McGurk stimuli (an auditory-/ba/-visual-/ga/ combination). Such changes in perception of clear (end-point) auditory stimuli following selective adaptation has been reported previously (e.g., Sawusch, 1976). This raises the question of whether the changes in perception reported across the two studies have anything at all to do with the visual-component of the McGurk stimuli, or if they are entirely driven by the auditory component of the McGurk stimuli. Since there was no auditory-alone training condition for either of the reported studies, it cannot be determined which part of the McGurk stimuli (the percept or the auditory component) is driving the changes in perception observed. This will have important implications for interpretation of the results as well as the validity of the CIMS model tested by the authors. I strongly encourage the authors to add another experiment to test whether training with only the auditory components of the McGurk stimuli results in changes in perception that are similar to those observed with their McGurk stimulus training. In lieu of such an effort, the discussion should include a thorough discussion of the possible role of selective adaptation in the observed results and what implications this will have for the CIMS model.

References

Cooper, W. E. (1974). Perceptuomotor adaptation to a speech feature. *Perception & Psychophysics*, 16(2), 229-234.

Cooper, W. E., & Lauritsen, M. R. (1974). Feature processing in the perception and production of speech. *Nature*, 252, 121-123.

Dias, J. W. (2016). Crossmodal Influences in Selective Speech Adaptation [University of California, Riverside]. <http://www.escholarship.org/uc/item/5sd725cp>

Dias, J. W., Cook, T. C., & Rosenblum, L. D. (2016). Influences of selective adaptation on perception of audiovisual speech. *Journal of Phonetics*, 56, 75-84. <https://doi.org/10.1016/j.jocn.2016.02.004>

Roberts, M., & Summerfield, Q. (1981). Audiovisual presentation demonstrates that selective adaptation in speech perception is purely auditory. *Perception & Psychophysics*, 30(4), 309-314.

Sawusch, J. R. (1976). Selective adaptation effects on end-point stimuli in a speech series. *Perception & Psychophysics*, 20(1), 61-65.

Reviewer #3 (Remarks to the Author):

I am happy with the changes

Response to Referees (Rebuttal letter)

We thank the three Referees for evaluating our resubmission, which contained additional data replicating our finding of fusion-induced recalibration (FIR) in the original stimulus from talker A.M. as well as another McGurk stimulus from a different talker composed of a different syllable pairing. Referees 1 and 3 were satisfied with the resubmission while Referee 2 raises the concern that our results could be explained by selective adaptation. As suggested by the Referee, we have added a new *Discussion* section on selective adaptation and made a number of changes to the text and figures (included a revised Figure 1C and the new Figure 2B) to highlight differences between FIR and selective adaptation.

Referee #1

I am satisfied with the authors' replies to my comments on the first manuscript. I think this is an interesting behavioral finding that contributes to a growing literature on multisensory recalibration. Congrats on a nice paper.

Referee #3

I am happy with the changes.

Referee #2

I greatly appreciate the authors' efforts to address the concerns raised by myself and the other reviewers in this revised manuscript. The expanded discussion and the addition of another experiment give greater theoretical grounding for the results and specificity to the findings. I find the manuscript to be greatly improved. I am, however, now greatly concerned that the results may be the result of something other than perceptual recalibration. A complimentary phenomenon to speech perception recalibration is the perceptual aftereffect known as "selective adaptation" (e.g., Cooper, 1974; Cooper & Lauritsen, 1974). In this phenomenon, perception of speech can be modulated by repeated presentation of within-modality speech stimuli. For example, perception of auditory speech stimuli along a /ba/-to-/va/ continuum can be modulated toward /ba/ (more stimuli along the continuum perceived as "va") following repeated presentation of an auditory /ba/ stimulus. Similarly, perception of auditory stimuli along the same /ba/-to-/va/ continuum can be modulated toward /va/ (more stimuli along the continuum perceived as "ba") following repeated presentation of an auditory "va" stimulus. Selective adaptation has famously been found to be strongest within-modality with very weak (if any) cross-sensory effects (Roberts & Summerfield, 1981) (see also Dias, 2016; Dias et al., 2016). What is observed in the studies reported in the current manuscript is described by the authors as a "recalibration" of the auditory stimuli toward the McGurk /da/ stimulus (auditory /ba/ toward perceived /da/). However, the results could also be described as a modulation of perception of the auditory /ba/ stimuli as less /ba/-like following "selective adaptation" to the auditory component of the McGurk stimuli (an auditory-/ba/-visual-/ga/ combination). Such changes in perception of clear (end-point) auditory stimuli following selective adaptation has been reported previously (e.g., Sawusch, 1976). This raises the question of whether the changes in perception reported across the two studies have anything at all to do with the visual-component of the McGurk stimuli, or if they are entirely driven by the auditory component of

the McGurk stimuli. Since there was no auditory-alone training condition for either of the reported studies, it cannot be determined which part of the McGurk stimuli (the percept or the auditory component) is driving the changes in perception observed. This will have important implications for interpretation of the results as well as the validity of the CIMS model tested by the authors. I strongly encourage the authors to add another experiment to test whether training with only the auditory components of the McGurk stimuli results in changes in perception that are similar to those observed with their McGurk stimulus training. In lieu of such an effort, the discussion should include a thorough discussion of the possible role of selective adaptation in the observed results and what implications this will have for the CIMS model.

Author Response:

The Referee proposes selective adaptation as an alternative explanation for FIR. As suggested by the Referee ("the discussion should include a thorough discussion of the possible role of selective adaptation in the observed results"), we have added a new *Discussion* section on selective adaptation and made a number of changes to the text and figures (included a revised Figure 1C and the new Figure 2B) to highlight differences between FIR and selective adaptation.

We agree with the Referee that selective adaptation is an interesting and important phenomenon. We thank the Referee for bringing the very interesting study of Dias *et al.*, 2016 to our attention and it is now described in the new *Discussion* section:

Other related perceptual phenomena: selective adaptation

Repeated presentations of auditory speech tokens can induce selective adaptation (Cooper, 1974; Cooper and Lauritsen, 1974; Eimas and Corbit, 1973; Sawusch, 1976). The typical procedure for selective adaptation is to present a continuum of synthetic speech stimuli that vary across some acoustic parameter. The stimuli at either ends of the continuum are in different phonetic categories, while the middle stimuli are ambiguous (perceived as one category on some trials and as another category on other trials). Following repeated exposure to an identical speech sound, the canonical finding is that perception of ambiguous stimuli shifts away from the adapted stimulus, as if (putative) feature detectors for that stimulus were fatigued by the repetition.

Selective adaptation has also been examined in the context of audiovisual speech perception (Dias *et al.*, 2016; Dorsi *et al.*, 2021; Roberts and Summerfield, 1981; Saldaña and Rosenblum, 1994; Samuel and Lieblich, 2014; Shigeno, 2002). Dias and colleagues developed a

clever variant of selective adaptation in which the test continuum was created by blurring the mouth region of the talker's face with progressively larger Gaussian filters (Dias et al., 2016). Auditory /ba/ paired with unblurred visual /va/ produced the percept of /va/ on 94% of trials, but this percept became less frequent (with a concomitant increase in the perception of /ba/) as blurring increased. Following adaptation to visual /va/ and audiovisual /va/, perception of the audiovisual continuum was modulated, but not following adaptation to auditory /va/, auditory /ba/, or visual /ba/, suggesting that auditory and visual speech information are not completely integrated at the level of selective adaptation.

There are profound differences between the circumstances in which selective adaptation and FIR are observed. The timescales of selective adaptation and FIR differ by orders of magnitude. Selective adaptation is observed for a few seconds following repetition of the adapting stimulus; after a few seconds, adaptation dissipates and additional "top-up" adapting stimuli must be presented. In contrast, FIR persists for days, weeks, or months, without any additional exposure to McGurk stimuli (*Figure 2A*). Selective adaptation requires many (dozens to hundreds) presentations of an adapting stimulus in quick succession (typical interstimulus intervals of ~500 ms). In contrast, FIR was induced with 5 daily repetitions of a McGurk stimulus, with each repetition spaced by several seconds. In selective adaptation, a continuum of synthetic speech stimuli, including ambiguous speech, are presented, while for FIR, the auditory speech always consisted of real talker speaking unambiguously.

Consistent with these major differences, no evidence for selective adaptation was observed in the present study. McGurk perceivers repeatedly experienced the fusion percept of *dada* on audiovisual exposure days. Under the adaptation account, this should decrease percepts of *dada* (as *dada* feature detectors grew fatigued). Instead, the opposite effect was observed, with

McGurk perceivers reporting many more *dada* percepts in the auditory-only post-test than in the pre-test. Auditory perceivers repeatedly experienced the auditory percept of *baba* on audiovisual exposure days. If selective adaptation was at play, a decrease in percepts of *baba* would be expected, but instead, percepts of *baba* remained at 100% in the post-test. These results are in line with the failure to observe any evidence for selective adaptation in a short-term study of phonetic recalibration (Lüttke et al., 2018).

The Referee suggests an experiment in which unambiguous auditory-only */baba/* is presented repeatedly to see if this leads to a shift in perception to percept of */dada/*. Data similar to this is shown in the new Figure 2B, reproduced here:

Participants were exposed to multiple repetitions of unambiguous auditory-only */baba/* on multiple days, and perception was completely stable: the percept of */dada/* was never reported.

FIR is remarkable precisely because it can lead to changes in the perception of unambiguous */baba/*, but **only subsequent to an error signal induced by repeatedly experiencing the McGurk effect**. In answer to the Referee's question about whether FIR has "anything at all to do with the visual-component of the McGurk stimuli, or if they are entirely driven by the auditory component of the McGurk stimuli", the revised Figure 1C is reproduced here:

The data shows the perception of unambiguous auditory */baba/*. The data in the left bar plot are from participants who were repeatedly exposed to unambiguous auditory */baba/* (paired with incongruent visual */gaga/*) but did not experience the McGurk effect. These participants showed no change in perception, and always perceived unambiguous auditory-only */baba/* veridically, as */baba/*. In contrast, the data in the right bar plot is from participants who *did* experience the McGurk effect. This induced a change in perception, so that unambiguous auditory-only */baba/* was perceived as */dada/* (FIR; red ellipse). Critically, both groups of participants were exposed to exactly the same number of unambiguous */baba/* stimuli (alone and as part of the McGurk pairing), providing additional evidence that perceiving the McGurk effect (rather than selective adaptation) drives FIR.

Response to Reviewers References

(see main manuscript for complete reference list)

13th Feb 24

Dear Professor Beauchamp,

Your manuscript titled "Fusion-Induced Recalibration: A long-lasting aftereffect of the McGurk effect" has now been seen by our reviewers, whose comments appear below. In light of their advice I am delighted to say that we are happy, in principle, to publish a suitably revised version in Communications Psychology under the open access CC BY license (Creative Commons Attribution v4.0 International License).

We therefore invite you to revise your paper one last time to address the remaining concerns of our reviewers and a list of editorial requests. At the same time we ask that you edit your manuscript to comply with our format requirements and to maximise the accessibility and therefore the impact of your work.

EDITORIAL REQUESTS:

SUBMISSION INFORMATION:

OPEN ACCESS:

Communications Psychology is a fully open access journal. Articles are made freely accessible on publication under a CC BY license (Creative Commons Attribution 4.0 International License). This license allows maximum dissemination and re-use of open access materials and is preferred by many research funding bodies.

For further information about article processing charges, open access funding, and advice and support from Nature Research, please visit <https://www.nature.com/commspsychol/article-processing-charges>

At acceptance, you will be provided with instructions for completing this CC BY license on behalf of all authors. This grants us the necessary permissions to publish your paper. Additionally, you will be asked to declare that all required third party permissions have been obtained, and to provide billing information in order to pay the article-processing charge (APC).

* TRANSPARENT PEER REVIEW: Communications Psychology uses a transparent peer review system.

On author request, confidential information and data can be removed from the published reviewer reports and rebuttal letters prior to publication. If you are concerned about the release of confidential data, please let us know specifically what information you would like to have removed. Please note that we cannot incorporate redactions for any other reasons.

* CODE AVAILABILITY: All Communications Psychology manuscripts must include a section titled "Code Availability" at the end of the methods section. We require that the custom analysis code supporting your conclusions is made available in a publicly accessible repository at this stage; please choose a repository that generates a digital object identifier (DOI) for the code; the link to the repository and the DOI must be included in the Code Availability statement. Publication as Supplementary Information will not suffice.

* DATA AVAILABILITY:

[link redacted]

Best regards,

Antonia Eisenkoeck

Antonia Eisenkoeck
Senior Editor
Communications Psychology

REVIEWERS' COMMENTS:

Reviewer #2 (Remarks to the Author):

I am satisfied with the changes made to the reporting of the data and the Discussion.